# The Link Between Sleep-Related Breathing Disorders and Idiopathic Pulmonary Fibrosis: Pathophysiological Mechanisms and Treatment Options—A Review

**DOI:** 10.3390/jcm14072205

**Published:** 2025-03-24

**Authors:** Athina Patsoura, Giulia Baldini, Daniele Puggioni, Matteo Delle Vergini, Ivana Castaniere, Dario Andrisani, Filippo Gozzi, Anna Valeria Samarelli, Giulia Raineri, Sofia Michelacci, Cristina Ruini, Andrea Carzoli, Aurelia Cuculo, Alessandro Marchioni, Bianca Beghè, Enrico Clini, Stefania Cerri, Roberto Tonelli

**Affiliations:** 1Respiratory Disease Unit, Department of Medical and Surgical Sciences of Motherhood and Child, University Hospital of Modena, 41125 Modena, Italy; athinapats27@gmail.com (A.P.); giulia.baldini.cr@gmail.com (G.B.); danielepuggionii@gmail.com (D.P.); 224231@studenti.unimore.it (M.D.V.); ivana_castaniere@icloud.com (I.C.); andrisanidario@gmail.com (D.A.); fillo.gzz@gmail.com (F.G.); sofiamichelacci97@gmail.com (S.M.); cristinaruini5@gmail.com (C.R.); andrea.carzoli@icloud.com (A.C.); aurcuculo@gmail.com (A.C.); marchioni.alessandro@unimore.it (A.M.); bianca.beghe@unimore.it (B.B.); stefania.cerri@unimore.it (S.C.); rtonelli@unimore.it (R.T.); 2Experimental Pneumology Laboratory, University of Modena and Reggio Emilia, 41125 Modena, Italy; annavaleria.samarelli@unimore.it (A.V.S.); giuliaraineri@unimore.it (G.R.); 3Center for Rare Lung Diseases, University Hospital of Modena, 41125 Modena, Italy

**Keywords:** sleep breathing disorders, idiopathic pulmonary fibrosis, obstructive sleep apnea, nocturnal hypoxemia, central sleep apnea, continuous positive airway pressure, high-flow nasal cannula, conventional oxygen therapy

## Abstract

In recent years, several studies have examined the impact of sleep-disordered breathing (SBD) on the quality of life and prognosis of patients with idiopathic pulmonary fibrosis (IPF). Among these disorders, obstructive sleep apnea (OSA) and nocturnal hypoxemia (NH) are the most prevalent and extensively studied, whereas central sleep apnea (CSA) has only been documented in recent research. The mechanisms underlying the relationship between IPF and SBDs are complex and remain an area of active investigation. Despite growing recognition of SBDs in IPF, no standardized guidelines exist for their management and treatment, particularly in a population characterized by distinct structural pulmonary abnormalities. This review outlines the pathophysiological connections between sleep-breathing disorders (SBDs) and idiopathic pulmonary fibrosis (IPF), as well as current therapeutic options. A comprehensive literature search using PubMed identified relevant studies, confirming the efficacy of CPAP in treating severe OSA and CSA. While high-flow oxygen therapy has not been validated in this patient cohort, it may offer a potential solution for select patients, particularly the elderly and those with low compliance. Conventional oxygen therapy, however, is limited to cases of isolated nocturnal hypoxemia or mild central sleep apnea.

## 1. Introduction

Idiopathic pulmonary fibrosis (IPF) is a chronic interstitial lung disease (ILD) marked by progressive fibrosis and represents the most prevalent form of idiopathic interstitial pneumonia. The disease is associated with a poor prognosis, with a median survival ranging from 3 to 5 years post-diagnosis [1,2]. IPF patients frequently present with various comorbidities that exacerbate prognosis and diminish quality of life [3].

Among these, sleep-related breathing disorders (SBDs) are particularly impactful, with obstructive sleep apnea (OSA) and nocturnal hypoxemia (NH) being the most common. Studies estimate the prevalence of OSA in IPF patients to range from approximately 64% to 88% [4,5,6,7,8], while NH, which may occur independently of OSA, affects about a quarter of IPF patients [9]. In contrast, fewer studies address the prevalence of central sleep apnea (CSA) in IPF. A recent study by Bordas-Martinez et al., involving 50 IPF patients with SBD, reported CSA in 12% of cases [10], while Hagmeyer et al. observed a 19% CSA prevalence in a cohort of 45 IPF patients [11]. Figure 1 illustrates IPF-related SBDs and the pathophysiological mechanisms most commonly associated with each of them.

Sleep disorders significantly impair sleep quality, further degrade overall quality of life, and correlate with poorer outcomes in IPF [12,13,14,15,16,17]. Prompt diagnosis and management of sleep disorders in IPF are essential for enhancing patients’ quality of life and survival. Our review is the first to comprehensively address all sleep-related respiratory disorders in this specific patient population, with an emphasis on the pathophysiological mechanisms connecting these disorders to IPF. Furthermore, we have provided an overview of the available therapeutic options for each disorder, although studies on this subject, particularly those concerning central apneas and nocturnal hypoxemia, are limited.

## 2. Sleep Architecture in IPF

Abnormalities in sleep architecture among IPF patients have been examined in multiple studies, revealing reduced sleep efficiency, decreased REM sleep and slow-wave sleep, shorter total sleep time, and increased stage 1 sleep (N1) and arousal index, contributing to sleep fragmentation [18,19,20]. These disturbances occur independently of OSA and do not appear to cause excessive daytime sleepiness, possibly due to the predominant fatigue experienced by IPF patients, which may overshadow daytime sleepiness [13]. Additionally, in the presence of concurrent pulmonary hypertension (PH), more pronounced sleep alterations have been reported, including increased wake time after sleep onset (WASO) and further reductions in sleep efficiency, likely driven by hypoxemia and heightened respiratory drive. These findings highlight the significant impact of sleep disturbances in IPF and related conditions, underscoring the need for further research to optimize management strategies [21]. Table 1 reports the most recent studies on sleep architecture in IPF patients.

## 3. IPF and Obstructive Sleep Apneas

The interplay between IPF and OSA is characterized by a complex, bidirectional relationship. IPF is defined by excessive extracellular matrix (ECM) protein deposition, leading to fibroblast activation and myofibroblast differentiation, which perpetuates a self-sustaining cycle of collagen accumulation. This process disrupts gas exchange, resulting in hypoxemia, increased lung stiffness, and decreased compliance [22,23].

OSA arises from both anatomical factors (visceral obesity, craniofacial abnormalities, airway collapsibility) and non-anatomical factors (high loop gain, low arousal threshold, reduced responsiveness of pharyngeal muscles), leading to upper airway collapse during sleep. OSA severity is quantified by the AHI index (>5 events/hour) [24,25]. While anatomical risk factors, particularly visceral obesity, play a central role in OSA pathogenesis, their correlation with OSA severity appears controversial in ILD patients. Several studies show that the degree of visceral obesity or neck circumference do not seem to correlate with the severity of OSA in ILD patients, suggesting a significant contribution of non-anatomical mechanisms [14,26]. However, from the study of Pereira et al., BMI emerges as the only independent predictor of OSA, indicating that excess weight could play a role in ILD patients [27]. A recent meta-analysis of Wei et al. confirms BMI as a risk factor linked to OSA in patients with IPF [28]. The reviewed studies suggest a co-dependent pathogenetic relationship between IPF and OSA, with the primary underlying mechanisms outlined below.

### 3.1. Recurring Alveolar Microdamage

The exact mechanism is not fully understood, but recurring alveolar microdamage has been observed during apnea episodes, where traction injury occurs at the alveolar level due to the necessity of breathing against a closed glottis. This process leads to significant intrathoracic and alveolar pressure fluctuations, resulting in epithelial damage and repeated traction lesions. It is possible that this mechanism contributes to fibrosis progression in ILD patients [29,30].

### 3.2. Intermittent Night-Time Hypoxemia and Systemic Inflammation

The desaturation–reoxygenation phases characteristic of OSA result in episodes of intermittent nocturnal hypoxemia, triggering systemic inflammation, oxidative stress, and vascular endothelial damage. Notably, intermittent hypoxemia induces a greater release of pro-inflammatory cytokines—including TNF-α, CRP, IL-6, and HIF-1—compared to continuous hypoxemia, which is typical of IPF. Shared risk factors between OSA and IPF further contribute to systemic inflammation through the release of pro-inflammatory mediators. Key risk factors include obesity, gastroesophageal reflux disease (GERD), and pulmonary hypertension. Studies have demonstrated increased leptin and HIF-1α production by adipocytes, elevated levels of TIMP-1, CINC1-2a/b, IL-1α, MIG-1, LIX, MIP1-3α, NF-κB, and PI-3k-Akt during pulmonary hypertension, and significant increases in LDH, ALP, CRP, and TNF-α in the bronchoalveolar lavage of GERD patients, all contributing to a systemic inflammatory state [31,32,33,34,35]. Specifically, GERD has been suggested to cause microaspirations that result in chronic lung parenchymal damage. Additionally, the increased inspiratory effort during OSA may reduce pleural pressure and increase transdiaphragmatic pressure, promoting GERD and perpetuating parenchymal injury [36]. Furthermore, intermittent nocturnal hypoxemia, intrathoracic pressure fluctuations, and sleep fragmentation associated with OSA promote catecholamine secretion, leading to arterial vasoconstriction and increased left ventricular afterload. This contributes to left ventricular diastolic dysfunction, ultimately elevating pulmonary arterial pressure and predisposing to pulmonary hypertension [37,38]. In addition, an experimental study in mice demonstrated that intermittent hypoxia induces Endoplasmic Reticulum (ER) stress, which plays a critical role in the development of IPF [39].

### 3.3. Vascular Alterations

Intermittent hypoxemia characteristic of patients with OSA is associated with endothelial dysfunction due to the initiation of inflammatory processes [31,32,33,34,35]. Pulmonary vascular involvement is further supported by Hagmeyer et al., who conducted cardiopulmonary exercise testing (CPET) in IPF patients with and without SBD. Among IPF patients with SBDs, functional impairments (e.g., reduced exercise capacity) and pulmonary vascular dysfunction (indicated by an elevated V’E/V’CO_2_ ratio, suggesting increased pulmonary vascular resistance) were significantly worse, highlighting a link between SBDs and vascular impairment. They propose that the intermittent hypoxemia occurring during OSA not only induces oxidative stress, leading to vascular remodeling, but also reduces peripheral chemoreceptor sensitivity, potentially exacerbating peripheral desaturation and delaying arousal from sleep. However, this finding contrasts with the perspective that hypoxemia, particularly if intermittent, may increase peripheral chemoreceptor sensitivity [11]. Trakada et al. also investigated pulmonary vascular involvement by measuring circulating endothelin-1 levels in ILD patients during sleep. Hypoxemia stimulates the release of endothelin-1, which induces vasoconstriction. They observed that circulating endothelin-1 levels were significantly elevated in ILD patients during sleep, especially in those with pulmonary hypertension, further illustrating the connection between ILD and OSA [40].

### 3.4. Reduction of Lung Volumes

Although data on these patients remain limited, Khor et al. suggest that the reduction in lung volumes and the fibrotic infiltration of lung parenchyma, characteristic of interstitial lung diseases (including IPF), diminish caudal tracheal traction on the upper airways. This reduction increases the pressure exerted by surrounding tissues on the upper airways, as well as the pressure required to close and reopen them. Additionally, decreased lung volumes lead to increased upper airway resistance. Collectively, these mechanisms contribute to enhanced pharyngeal collapse, promoting apnea, particularly during REM sleep [41,42]. To confirm, a study conducted in a cohort of non-IPF fibrotic ILD patients found that a lower TLC was correlated with a higher risk of OSA. Further studies in ILD patients are needed to fully elucidate this mechanism [43].

### 3.5. High Loop Gain

Loop gain, a key determinant of ventilatory control stability, is one of the primary non-anatomical risk factors in the development of OSA [44]. It is influenced by two components: the chemosensitivity of peripheral and central receptors to hypoxemia and hypercapnia (controller gain) and the impact of ventilation on gas exchange (plant gain). During apnea, airflow and lung volumes decrease, leading to hypoventilation [45]. This results in altered blood gases (increased PaCO_2_ and decreased PaO_2_), which activate chemoreceptors (controller gain) and trigger a compensatory ventilatory response characterized by hyperventilation and a subsequent reduction in PaCO_2_ (plant gain). Intermittent hypoxemia, a hallmark of patients with OSA, enhances the sensitivity of chemoreceptors, particularly the peripheral ones. This heightened sensitivity results in an increased ventilatory response, which is positively correlated with controller gain [25]. Plant gain instead is influenced by two main factors: reduced lung volumes and hypercapnia. Reduced lung volumes, commonly observed in OSA (particularly the decreased functional residual capacity, or FRC, which is often associated with obesity in these patients), are linked to a diminished buffering capacity of the lungs for PaCO_2_. This condition also contributes to increased airway collapse, further amplifying ventilatory instability and thus plant gain [44]. Furthermore, the increase in PaCO_2_ levels during apnea, resulting from hypoventilation, can provoke an exaggerated ventilatory response. Rapid and abrupt fluctuations in PaCO_2_ exacerbate ventilatory instability, thereby increasing plant gain [25]. Although no studies have specifically investigated loop gain in IPF patients, Khor et al. suggest that the reduced lung volumes and impaired gas exchange—particularly, persistent hypoxemia—characteristic of these patients may contribute to increased plant gain and controller gain, thereby elevating loop gain [42].

### 3.6. Low Arousal Threshold

The arousal threshold represents the stimulus intensity required to trigger awakening and constitutes a critical non-anatomical risk factor for OSA. In OSA patients, this threshold is significantly reduced, leading to heightened arousability and sleep fragmentation—a phenomenon similarly observed in ILD patients. In ILD, the chronic elevation of ventilatory drive, driven by increased work of breathing and impaired gas exchange, imposes substantial physiological constraints. During OSA episodes, the ventilatory reserve in these patients is markedly reduced, limiting their capacity to further augment ventilatory effort as non-ILD individuals do during sleep. Consequently, this reduced reserve facilitates the surpassing of the arousal threshold, contributing to an elevated arousal index. However, further investigation is required to elucidate the precise interplay between the arousal threshold, work of breathing, and OSA pathophysiology in ILD patients, particularly those with IPF [46].

## 4. IPF and Nocturnal Hypoxemia

NH is highly prevalent in IPF patients and frequently remains undiagnosed. Significant nocturnal desaturation (SND), defined as ≥10% of total sleep time with oxygen saturation ≤90%, can occur even in the absence of OSA [20]. The pathophysiological mechanisms underlying NH in IPF include hypoventilation during REM sleep due to diminished skeletal muscle activity and a reduction in functional residual capacity in the supine position [47], as well as ventilation–perfusion mismatch resulting from interstitial lung disease and increased pulmonary physiologic dead space [48]. Coexisting respiratory disorders, such as OSA and central sleep apnea (CSA), further exacerbate nocturnal desaturation. Yasuda et al. analyzed nocturnal desaturation patterns in IPF patients and identified sustained desaturation—defined as a decline in SpO_2_ > 3% from baseline lasting more than 655 s—as the pattern associated with the lowest SpO_2_ levels and the highest proportion of total sleep time spent with SpO_2_ < 90%. The intermittent desaturation pattern, characterized by recurring cycles of SpO_2_ decline and recovery over several minutes, was linked to OSA, whereas the periodic pattern, in which SpO_2_ drops lasted less than 655 s, showed no significant correlation with sleep parameters. Classifying desaturation waveforms may provide insight into the severity of hypoxemia and potential complications related to OSA [49,50]. Several studies have demonstrated that NH is associated with poor prognosis in IPF patients. Prolonged nocturnal hypoxemia, rather than OSA alone, has been linked to accelerated deterioration in quality of life and an increased risk of all-cause mortality in patients with fibrotic interstitial lung disease [51]. In the study by Troy et al., SND occurred independently of OSA and was predictive of both survival and the development of pulmonary hypertension at 12 months, as assessed by echocardiography [9]. Similarly, Margaritopoulos et al. reported that nocturnal desaturation in ILD patients correlated with non-invasive markers of pulmonary hypertension, including tricuspid regurgitation velocity, brain natriuretic peptide levels, carbon monoxide transfer coefficient, alveolar–arterial oxygen gradient, desaturation >4% during the 6-min walk test, and pulmonary artery diameter ≥29 mm. These findings suggest a strong association between NH and pulmonary vasculopathy [52]. Furthermore, NH may contribute to the progression of pulmonary fibrosis through mechanisms such as alveolar collapse, ventilator-induced lung injury, and altered surfactant production [53,54]. Bosi et al. demonstrated that IPF progression was more severe in patients with both OSA and NH compared to those with only IPF and OSA, further underscoring the clinical significance of nocturnal hypoxemia in disease progression [14].

## 5. IPF and Central Sleep Apneas

While CSA can occur in patients with IPF, there is a notable lack of studies specifically investigating their prevalence and pathophysiological mechanisms in this population. CSA is characterized by the temporary cessation of ventilatory effort for at least 10 s during sleep, often accompanied by impaired gas exchange and dysregulation of respiratory drive [54]. In IPF, increased lung stiffness leads to reduced lung volumes, resulting in low tidal volume, decreased functional residual capacity, and mechanoreceptor activation, which subsequently heightens respiratory drive [55]. Additionally, gas exchange abnormalities—driven by interstitial vascular changes and alveolar–capillary membrane dysfunction—lead to a reduced diffusing capacity and chronic hypoxemia [56]. Persistent alterations in lung mechanics and gas exchange are associated with sustained activation of the central ventilatory command. In an animal model study, Yegen et al. demonstrated that mechanical changes and chronic hypoxemia can induce neuroplasticity in respiratory centers, leading to long-term functional modifications [57]. It can be hypothesized that such neuroplastic changes promote instability in the respiratory center, contributing to an elevated loop gain, which may underlie the development of central apneas in patients with pulmonary fibrosis. Additionally, ventilatory instability in IPF may be influenced by altered sensitivity of peripheral chemoreceptors to CO_2_ and O_2_ levels. Hyperventilation, a common feature in IPF, may reduce the CO_2_ reserve, lowering CO_2_ levels below the apnea threshold and thereby triggering central apneas [58]. Intermittent hypoxemia, frequently observed in the presence of coexisting OSA, may further exacerbate ventilatory instability by increasing chemosensitivity to low oxygen levels, thereby elevating controller gain and promoting central apneas [59]. In severe cases of IPF, pulmonary hypertension and progressive hyperventilation may further contribute to the occurrence of central apneas [60]. Although periodic breathing has not been described in IPF, a hyperpnea–hypopnea breathing pattern has been reported in patients with both ILD and OSA but not in those with OSA alone. This pattern is likely associated with central respiratory events. Notably, oxygen therapy has been shown to normalize these abnormal breathing patterns, suggesting a potential therapeutic role in the management of central apneas in IPF patients [61].

## 6. IPF Associated with Emphysema and SBD

According to the literature, there is increasing recognition of the overlap syndrome between COPD (chronic obstructive pulmonary disease) and OSA (obstructive sleep apnea) [62]. IPF is frequently associated with emphysema, predominantly affecting the upper lobes, within the context of CPFE (combined pulmonary fibrosis and emphysema) syndrome [63]. While emphysema is known to share a well-established pathophysiological link with COPD, evidence supporting an association between CPFE and sleep-breathing disorders (SBD) remains limited. However, in two studies that assessed the characteristics of patients with CPFE by comparing them to those with isolated IPF, the CPFE group demonstrated a comparable prevalence of obstructive sleep apnea [64,65].

## 7. Biomarkers

Currently, limited evidence exists in the literature regarding biomarkers common to both IPF and OSA, despite numerous studies investigating each disease individually [66]. An ideal biomarker should be specific, sensitive, clinically relevant, and cost-effective. In OSA, biomarker expression is influenced by systemic inflammation due to sleep deprivation, hypoxemia, obesity, parasympathetic nervous system activation during apneic episodes, comorbid diabetes mellitus, and increased reactive oxygen species (ROS) production secondary to nocturnal hypoxemia [67,68]. Emerging data suggest a potential link between increased plasma levels of CCL18 in IPF and nocturnal hypoventilation [68,69]. Additionally, increased levels of ET-1, KL-6, and S100A9 calcium-binding protein [70,71,72,73,74,75,76] have been found to be elevated in both IPF and OSA, suggesting shared mechanisms underlying fibrosis development, though further investigation is required. Epigenetic regulation, particularly DNA methylation, is an area of growing interest in both diseases. In OSA, DNA methylation of the FOXP3 gene has been linked to inflammatory responses and clinical disease severity in pediatric patients. Similarly, microRNA (miRNA) expression plays a pivotal role in IPF pathogenesis; however, direct epigenetic correlations between OSA and IPF remain unclear. Identifying specific miRNA signatures in OSA may provide insight into novel pathogenic mechanisms and potential therapeutic targets [77,78].

## 8. IPF and SBDs: Treatment Options

### 8.1. Continuous Positive Airway Pressure

Continuous positive airway pressure (CPAP) remains the gold standard treatment for OSA, as it effectively alleviates upper airway obstruction, improves nocturnal oxygenation, and enhances sleep quality [79,80]. However, the impact of mechanical ventilation (MV) in IPF patients remains poorly understood, with potential concerns regarding its role in fibrosis progression. Mechanical stress may contribute to the dysregulation of key molecular pathways involved in lung tissue repair, potentially promoting fibrotic changes [81].

Preliminary data from patients with acute exacerbations of ILD undergoing MV—both invasive and non-invasive—demonstrate a notable mechanical disadvantage, particularly in the context of high positive end-expiratory pressure (PEEP). This phenomenon has been explained using the “lung squishy-ball” model, which suggests that fibrotic lungs are susceptible to overdistension-related injury when subjected to excessive ventilatory pressures [82,83]. However, these observations are derived from patients requiring MV during acute respiratory failure, rather than under baseline conditions. Given these considerations, the potential role of CPAP in IPF patients with moderate-to-severe OSA (AHI ≥ 15 events/hour) was evaluated in several studies [84,85]. Five relevant articles were identified [86,87,88,89,90], with one excluded due to its broader focus on interstitial lung diseases [88]. The first study, conducted over 6 months in 12 patients, was the first to demonstrate a significant improvement in quality of life with CPAP therapy, assessed using the Functional Outcomes of Sleep Questionnaire (FOSQ), a validated tool measuring the impact of sleep disorders on daily activities [86]. A subsequent study evaluated 92 newly diagnosed IPF patients who underwent polysomnography (PSG). Among them, 45 patients (49%) initiated CPAP therapy and were stratified into good and poor CPAP compliance groups. After 1 year, the good-compliance group (*n* = 37) showed significant improvements across all quality-of-life and sleep-related metrics, whereas the poor-compliance group (*n* = 18) exhibited smaller, yet still significant, improvements in only a subset of these measures. Over 24 months, three patients from the poor-compliance group died, while all patients in the good-compliance group survived, suggesting that consistent CPAP use may confer a survival benefit. It is important to note that these studies were conducted before the widespread use of antifibrotic therapies, during which disease progression was typically more rapid and prognosis poorer [87]. Another study by Papadogiannis et al. (2021) [12] assessed 45 patients, 29 of whom received positive airway pressure (PAP) therapy. At the 7-year follow-up, CPAP therapy was associated with improvements in sleepiness, fatigue, and sleep quality. Patients with CPAP adherence of <4 h per night demonstrated less pronounced quality-of-life improvements, while those with ≥4 h of use experienced more substantial benefits. Furthermore, patients with ≥6 h of nightly CPAP adherence exhibited better survival compared to those using CPAP for <6 h per night [12]. However, this study does not provide data on the prevalence of pulmonary hypertension, which may be associated with more severe forms of IPF and negatively affect patient outcomes. In these patients, the effects of PAP therapy on sleepiness, fatigue, and survival may differ compared to those with mild forms of IPF [89]. More recently, Bordas-Martinez et al. enrolled 50 IPF patients screened for SBDs, reporting a 70% prevalence of SBDs, including OSA (32%), CSA (22%), and sleep-sustained hypoxemia (SSH) (12%). CPAP therapy was initiated in all OSA patients and in CSA patients whose respiratory events were reduced by >50% following manual titration. After 1 year, CPAP adherence was optimal (6.74 h/night), yet no significant changes in pulmonary function or quality of life were observed. However, systemic markers, including MMP-1 (a profibrotic metalloproteinase associated with hypoxia), showed a significant reduction after 1 year of treatment. Notably, one-third of patients without baseline SBDs developed it over the year, necessitating treatment initiation, while 17% of those initially treated required increased CPAP pressure adjustments despite no functional deterioration or weight gain [90]. Pneumothorax is a potential complication of IPF [91,92,93], particularly in cases associated with emphysema (CPFE). Although it is plausible to hypothesize that CPAP therapy might increase the risk of pneumothorax through mechanisms of barotrauma, none of the patients included in the studies reviewed developed this complication, as well as there are only five reported cases of pneumothorax in association with Non-Invasive Positive Pressure Ventilation therapy for OSA in non-IPF patients, and none of them involved patient with interstitial lung disorder [94,95,96,97,98] (one of them had pan acinar emphysema and was treated with Bilevel–Positive Airway Pressure [95]. Based on these studies, CPAP remains the most widely used therapy for moderate-to-severe OSA in IPF patients, effectively reducing AHI and improving quality of life. However, the available data are derived from a limited number of patients with relatively short follow-up periods, limiting the ability to fully assess the long-term effects of mechanical ventilation in fibrotic lung disease. To minimize potential adverse effects on lung parenchymal mechanics, CPAP pressures should be titrated to the lowest effective level necessary to maintain upper airway patency while avoiding excessive parenchymal stretch, which could theoretically exacerbate fibrosis progression. Table 2 summarizes the key studies assessing CPAP therapy for OSA in patients with IPF.

Further studies with larger patient cohorts and extended follow-up periods are required to clarify the long-term impact of CPAP therapy in this population. Additionally, other ventilatory modalities, such as bilevel positive airway pressure (BiPAP) and adaptive servo-ventilation (ASV), which are employed in cases of persistent CSA despite CPAP therapy—particularly in patients with CSA due to heart failure, opioid-induced sleep apnea, and brainstem disorders—have not yet been systematically investigated in IPF patients with CSA [99].

### 8.2. High-Flow Nasal Cannula

High-flow nasal cannula (HFNC) has been investigated as a treatment modality for various respiratory conditions, including OSA and IPF. By increasing nasopharyngeal gas flow, HFNC effectively reduces overall dead space and the resistive work of breathing by matching or exceeding inspiratory resistance, thereby enhancing alveolar ventilation and gas exchange. Additionally, warmed and humidified air improves lung compliance and decreases the metabolic cost of conditioning nasopharyngeal gases. The generation of continuous distending pressure, reaching up to 7 cm H_2_O in the lungs, further enhances lung compliance by maintaining alveolar patency. However, HFNC efficacy is influenced by several factors, including flow rate, the flow-to-body-weight ratio, nasal cannula size, and mouth position [100,101,102]. In OSA, HFNC stabilizes breathing and reduces respiratory events during sleep by improving upper airway patency and ventilation. Due to its superior compliance in pediatric patients compared to CPAP, initial studies have primarily focused on this population. A recent meta-analysis by Du et al. concluded that HFNC significantly reduces the AHI and increases minimum oxygen saturation during sleep [103]. Similarly, Fischman et al., in a randomized trial of obese pediatric patients with moderate-to-severe OSA, demonstrated that HFNC achieved AHI reductions comparable to CPAP [104]. In adults, Yen et al. found that HFNC effectively decreased respiratory events, particularly in mild-to-moderate OSA, with greater benefits observed during REM sleep and in older individuals. This effect may be attributed to the physiological reduction in muscle tone during REM sleep and aging-associated neuromuscular changes [105]. Similarly, Yu et al. compared HFNC and CPAP in OSA and reported that HFNC reduced AHI, nighttime desaturation, and snoring while increasing minimum SpO_2_ levels. However, HFNC was also associated with an increase in central apneas, particularly in the supine position and during NREM sleep, and did not improve sleep quality to the same extent as CPAP [106]. Currently, no studies have evaluated HFNC efficacy in managing OSA in patients with IPF. However, HFNC has been validated for the management of acute respiratory failure during ILD exacerbations, demonstrating improved patient tolerability and a lower rate of therapy discontinuation compared to conventional oxygen therapy or non-invasive ventilation (NIV) [107]. Evidence suggests that HFNC reduces respiratory rate, alleviates dyspnea severity, and is non-inferior to NIV in ARF management in ILD patients [108]. Notably, the FLORALI trial demonstrated that HFNC significantly reduced 90-day mortality compared to NIV in this patient population [109]. Moreover, HFNC has been associated with improved exercise tolerance, reduced fatigue, and enhanced oxygenation during physical activity in IPF patients [110,111]. Despite its benefits, HFNC—similar to CPAP—can induce treatment-emergent central apneas by lowering PCO_2_ levels below the apneic threshold and activating the Hering–Breuer reflex due to increased lung volumes from distending pressure [112,113]. Given the improved tolerability of HFNC, further research is warranted to assess its efficacy in reducing apneic events in IPF patients and its potential role as an alternative to CPAP in this population.

### 8.3. Conventional Oxygen Therapy

Intermittent hypoxemia during sleep is a direct consequence of SBDs and is associated with an increased risk of cardiovascular and neurological complications. Oxygen supplementation mitigates intermittent hypoxemia, leading to a reduction in chemoreceptor sensitivity and, consequently, a decrease in controller gain. Additionally, hyperoxemia can stimulate hyperventilation, reducing plant gain through mechanisms such as increased brain tissue PCO_2_ due to cerebral vasoconstriction, the Haldane effect, or direct stimulation of central chemoreceptors. As a result, the CO_2_ reserve increases, promoting ventilatory stability and sleep consolidation [114,115]. A meta-analysis by Mehta et al. found that conventional oxygen therapy (COT) improves oxygen saturation in patients with OSA to a degree comparable to CPAP but does not reduce AHI or alleviate daytime sleepiness. However, in a series of six observational studies, oxygen therapy was associated with a significant reduction in apneic events compared to room air alone [116]. Patients with OSA who derive the greatest benefit from oxygen supplementation typically exhibit higher loop gain and lower upper airway collapsibility [117]. The effects of oxygen therapy on CSA in ILD patients remain unclear due to a lack of targeted research. However, COT has been shown to reduce AHI in patients with congestive heart failure, a population that may share similar pathophysiological mechanisms with IPF, including reduced CO_2_ reserve and altered chemoreceptor sensitivity—both of which may be stabilized with oxygen supplementation [118]. In ILD, NH is associated with reduced diffusion capacity for carbon monoxide (DLCO) and echocardiographic markers of pulmonary hypertension, both of which correlate with poorer survival outcomes. Despite these associations, studies evaluating the impact of oxygen supplementation in this population remain limited [119]. A high-altitude study demonstrated that ILD patients receiving 1–3 L/min of oxygen experienced improved nocturnal oxygen saturation and reductions in both heart and respiratory rates, achieving physiological parameters comparable to those of healthy controls [120]. Additionally, Al Aghbari et al. reported that supplemental oxygen in ILD patients with mild-to-moderate nocturnal hypoxia decreased the duration and frequency of desaturation events, thereby improving sleep-disordered breathing indices. However, sleep quality and architecture remained unchanged [121]. Further investigation is required to elucidate the role of oxygen therapy in stabilizing ventilatory control and mitigating the long-term consequences of nocturnal hypoxemia in patients with ILD and coexisting SBDs. Figure 2 shows the main treatment options for sleep breathing disorders in patients with IPF.

## 9. Future Perspectives

### 9.1. Artificial Intelligence

Artificial intelligence (AI) has demonstrated significant utility in the context of OSA, particularly for screening and treatment selection, enhancing the identification of at-risk patients, and facilitating personalized therapeutic strategies. Molnár et al. reported that AI-driven analysis of anthropometric (BMI, craniofacial characteristics), demographic (age, gender), and questionnaire-based data (e.g., Berlin and Epworth scales) improves high-risk detection, potentially reducing reliance on polysomnography for initial diagnosis [122,123]. Beyond screening, AI has shown potential in optimizing treatment selection by identifying underlying pathogenic mechanisms and enabling targeted interventions. Zhu et al. and Cederberg et al. demonstrated that AI algorithms can detect immune-activation patterns and biomarkers associated with OSA severity, with measurable improvements following CPAP therapy [124,125]. Despite these advancements, AI applications in OSA–ILD patients remain largely unexplored. Further research is warranted to assess AI’s potential in refining diagnostic accuracy, predicting disease progression, and tailoring therapeutic approaches in this population.

### 9.2. Radiological Features

Future research should further investigate the radiological correlation between ILD and OSA, as several studies have already explored this association. Notably, Kim et al. demonstrated that individuals with an AHI > 15 (indicative of at least moderate OSA) exhibited an 11.30% increase in high attenuation areas (HAAs, defined as regions with attenuation values between −600 and −250 Hounsfield units) and a 3.63% reduction in FVC, along with a 220.2 mL decline in TLC over 10 years, compared to individuals with an AHI < 5. Notably, BMI did not influence these associations [126]. These findings reinforce the link between OSA and interstitial lung diseases, underscoring the potential for further research aimed at integrating radiological and clinical data to improve early detection and risk stratification. The development of AI-driven models for automated screening and prognostic assessment in this patient population could enhance early diagnosis and facilitate personalized therapeutic approaches.

## 10. Conclusions

The prevalence of SBDs in patients with IPF is substantial and is associated with impaired quality of life, underscoring the need for systematic screening at diagnosis and throughout follow-up. The interplay between SBDs and IPF is driven by complex pathophysiological mechanisms, reflecting the multifaceted nature of this comorbidity. While therapeutic interventions such as CPAP, HFNC, and COT are available, their efficacy and optimal application in IPF patients remain insufficiently supported by high-quality evidence. SBDs in the context of IPF represent a largely underexplored domain, highlighting the urgent need for well-designed prospective studies to elucidate its pathophysiology, assess treatment efficacy, and refine clinical management strategies.

## Figures and Tables

**Figure 1 jcm-14-02205-f001:**
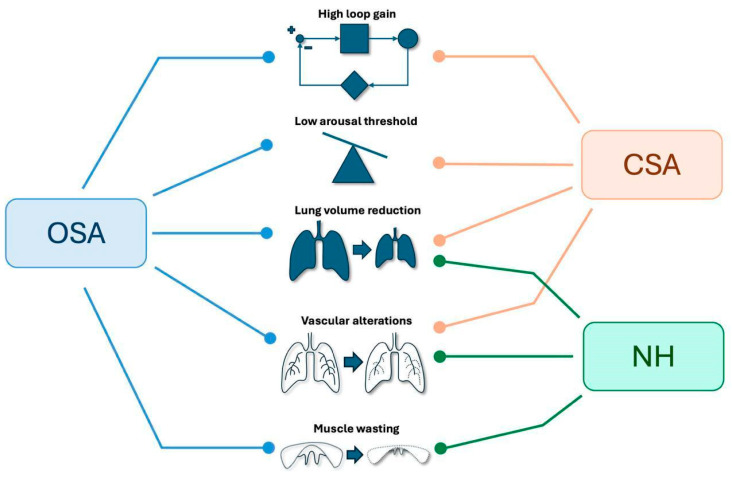
Primary SBDs identified in patients with IPF and the pathophysiological mechanisms underlying their development. SBDs: Sleep Breathing Disorders IPF: Idiopathic Pulmonary Fibrosis, OSA: Obstructive Sleep Apnea, CSA: Central Sleep Apnea, NH: Nocturnal Hypoxemia.

**Figure 2 jcm-14-02205-f002:**
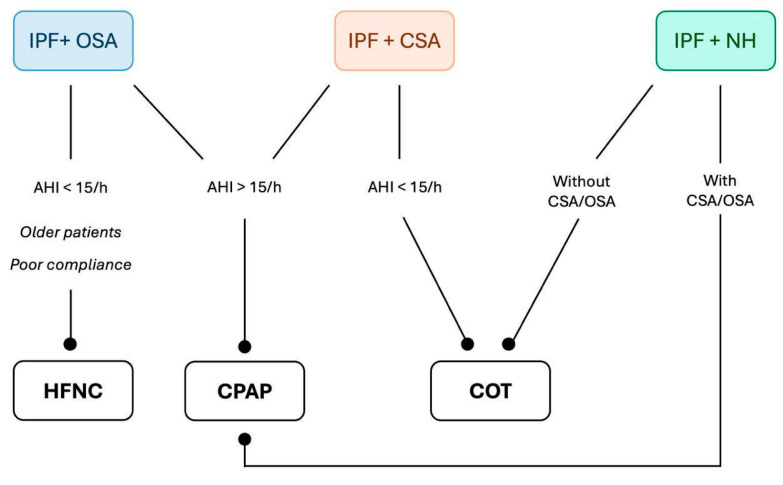
Proposal for a flowchart of treatment options for different sleep breathing disorders in patients with IPF. IPF: Idiopathic Pulmonary Fibrosis, OSA: Obstructive Sleep Apnea, CSA: Central Sleep Apnea, NH: Nocturnal Hypoxemia, HFNC: High-Flow Nasal Cannula, CPAP: Continuous Positive Airway Pressure, COT: Conventional Oxygen Therapy.

**Table 1 jcm-14-02205-t001:** Disturbances in sleep architecture in patients with IPF.

Author	Cohort	Modifications in Sleep Organization	Limitations
Mermigkis et al. 2007 [18]	18 patients with IPF	Reduction of sleep efficiency, REM sleep and slow-wave sleep, and increased arousal index. AHI positively correlated with BMI and negatively correlated with FEV1.	small-sized simple retrospective study
Mermigkis et al. 2009 [19]	15 patients with IPF compared with 15 controls	Decrease in sleep efficiency and slow-wave sleep. Increase in N1 sleep and arousal index. Daytime tachypnea persisted during sleep in patients with IPF.	small-sized simple patients not receiving any treatment excluding the advanced forms
Lancaster et al. 2009 [6]	50 patients with IPF	Reduction of sleep efficiency, slow-wave sleep, and REM sleep. Increased arousals, which correlated with the severity of OSA.	unselected sample: patients with prior diagnosis of OSA were included lack of a control group
Mermigkis et al. 2010 [20]	34 patients with IPF treatment naive	Decreased sleep efficiency and REM sleep. Increased N1 stage and arousal index.	small-sized simple lack of a control group patients not receiving any treatment excluding the advanced forms
Kolilekas et al. 2013 [13]	31 patients with IPF treatment naive	Decreased REM sleep. Increased arousal index and N2 stage. Sleep oxygen desaturation indices correlated with the right ventricular systolic pressure providing the link between intermittent oxygen desaturation and PH.	limited-sized simple lack of a control group
Simonson et al. 2022 [21]	24 patients with ILD and PH compared with 25 patients with ILD without PH	Reduced total sleep time, a decreased proportion of stage N2. Increased proportion of stage N1 and wake after sleep onset (WASO) and poor sleep quality.	small-sized sample retrospective study lack of data measuring arousals

**Table 2 jcm-14-02205-t002:** Impact of CPAP therapy in IPF patients with Obstructive Sleep Apnea.

Author	Cohort	Intervention	Main Findings	Conclusions
Mermigkis et al. 2013 [86]	12 IPF patients	CPAP for 6 months	Significant improvement in quality of life (measured by FOSQ)	First study demonstrating CPAP benefit in IPF patients
Mermigkis et al. 2015 [87]	92 IPF patients (45 on CPAP)	CPAP, stratified by compliance	Good compliance group (*n* = 37) had significant QOL and sleep improvement, survival benefit over 24 months	Consistent CPAP use may improve survival
Papadogiannis et al. 2021 [12]	45 IPF patients (29 on CPAP)	CPAP, adherence stratified	Improvements in sleepiness, fatigue, sleep quality; better survival with ≥6 h adherence	CPAP adherence impacts both QoL and survival
Bordas-Martinez et al. 2024 [90]	50 IPFpatients	CPAP	Optimal adherence (6.74 h/night); no significant pulmonary or QoL changes, but reduced MMP-1 levels	CPAP may reduce profibrotic markers

CPAP = Continuous Positive Airway Pressure; IPF = Idiopathic Pulmonary Fibrosis; QoL = Quality of Life; FOSQ = Functional Outcomes of Sleep; SBD = Sleep Breathing Disorders; MMP-1 = Matrix Metalloproteinase-1.

## Data Availability

No new data were created or analyzed in this study. Data sharing is not applicable to this article.

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
