# Peer review of "The Link Between Sleep-Related Breathing Disorders and Idiopathic Pulmonary Fibrosis: Pathophysiological Mechanisms and Treatment Options—A Review"

_jcm, 2025, doi:10.3390/jcm14072205_

Round 1
Reviewer 1 Report
Comments and Suggestions for Authors
The authors reviewed the clinical relevance of sleep-disordered breathing (SDB) in patients with idiopathic pulmonary fibrosis (IPF). They comprehensively described the epidemiology, pathophysiology, and treatment of SDB in patients with IPF. Although the manuscript was well structured and very informative, I found some points to address in addition to the original one. My comments are as follows:
Major comments
- Although the publications may be limited, the anatomical factors, such as obesity, craniofacial deformity and airway collapse, in patients with IPF should be addressed in a more detailed fashion.
- Some clinicians are concerned with the potential risk for pneumothorax when patients with IPF receive continuous positive airway pressure (CPAP) ventilation. I’m not not sure whether CPAP can really increase the risk for pneumothorax in patients with IPF or cystic lung lesions. IPF is an established underlying disease for secondary pneumothorax. The authors should address the potential link between CPAP and pneumothorax in patients with parenchymal lung diseases, such as IPF and emphysema.
- Some patients with IPF have concomitant emphysema, designated as combined pulmonary fibrosis and emphysema (Cottin V, et al. Am J Respir Crit Care Med 2022). Obstructive sleep apnea (OSA) is an increasingly recognized comorbidity in patients with chronic obstructive pulmonary disease, which is known as COPD-OSA overlap syndrome (Srivali N, et al. Sleep Med 2023). The potential effects of concomitant emphysema or COPD on the relationships between IPF and SDB should be addressed, if prior studies about this issue are available.
Minor comment
- It is difficult to understand the reason why reduced lung volumes and impaired gas exchange can increase plant gain. To me, it seems that reduced lung volumes and impaired gas exchange may complicate the ventilation and thus prevent plant gain.
Author Response
Reviewer 1
The authors reviewed the clinical relevance of sleep-disordered breathing (SDB) in patients with idiopathic pulmonary fibrosis (IPF). They comprehensively described the epidemiology, pathophysiology, and treatment of SDB in patients with IPF. Although the manuscript was well structured and very informative, I found some points to address in addition to the original one. My comments are as follows:
Major comments
- Although the publications may be limited, the anatomical factors, such as obesity, craniofacial deformity and airway collapse, in patients with IPF should be addressed in a more detailed fashion.
Response to Reviewer 1’s major comment 1
We sincerely thank the Reviewer for the accurate reviewing process of our manuscript and for the appreciation of our work. We haven’t explored in depth the relationship between the OSA anatomical factors in patients with OSA and IPF because most of the studies that we found in literature, don’t correlate OSA in IPF patients to anatomical factors but to non-anatomical factors. However, we have modified the paragraph taking your comment into account.
- Some clinicians are concerned with the potential risk for pneumothorax when patients with IPF receive continuous positive airway pressure (CPAP) ventilation. I’m not not sure whether CPAP can really increase the risk for pneumothorax in patients with IPF or cystic lung lesions. IPF is an established underlying disease for secondary pneumothorax. The authors should address the potential link between CPAP and pneumothorax in patients with parenchymal lung diseases, such as IPF and emphysema.
Response to Reviewer 1’s major comment 2:
We thank the reviewer for her/his comment. IPF patients, as well as those with emphysema, are surely exposed to a higher risk of spontaneous pneumothorax compared to the healthy population. However, we didn’t find strong evidence of increased risk of pneumothorax during somministration of continuous positive airways pressure. We have now clarified that in the section dedicated to treatments with CPAP.
- Some patients with IPF have concomitant emphysema, designated as combined pulmonary fibrosis and emphysema (Cottin V, et al. Am J Respir Crit Care Med 2022). Obstructive sleep apnea (OSA) is an increasingly recognized comorbidity in patients with chronic obstructive pulmonary disease, which is known as COPD-OSA overlap syndrome (Srivali N, et al. Sleep Med 2023). The potential effects of concomitant emphysema or COPD on the relationships between IPF and SDB should be addressed, if prior studies about this issue are available.
Response to Reviewer 1’s major comment 3:
We thank the reviewer for her/his comment. The topics addressed are undoubtedly of interest and merit consideration; however, we were unable to identify sufficient evidence in the literature to establish a clear association between these conditions. We have added a mention of this in the text.
Minor comment
- It is difficult to understand the reason why reduced lung volumes and impaired gas exchange can increase plant gain. To me, it seems that reduced lung volumes and impaired gas exchange may complicate the ventilation and thus prevent plant gain.
Response to Reviewer 1’s minor comment 1
We thank the reviewer for her/his comment and we do apologize if the paragraph wasn’t clear. We have now expanded the paragraph to explain better how reduced lung volumes and impaired gas exchange can influence plant gain.
Reviewer 2 Report
Comments and Suggestions for Authors
This manuscript is a narrative review to evaluate the impact of sleep-disordered breathing (SBD) on the quality of life and prognosis of patients with idiopathic pulmonary fibrosis (IPF). Among these disorders, obstructive sleep apnea (OSA) and nocturnal hypoxemia (NH) are the most prevalent and extensively studied, whereas central sleep apnea (CSA) has only been documented in recent research. This review provides an overview of the pathophysiological correlations between SBDs and IPF, as well as the currently available therapeutic options, emphasizing the need for further research to optimize clinical management strategies.
It is an interesting work.
It presents excellent figures to show their findings.
I recommend its publication after some minor improvements:
- The authors should describe what this manuscript adds to previous reviews already published: ref 3, 4, etc.
- Comment on: Wei CR, Jalali I, Singh J, et al. Exploring the Prevalence and Characteristics of Obstructive Sleep Apnea Among Idiopathic Pulmonary Fibrosis Patients: A Systematic Review and Meta-Analysis. Cureus. 2024;16(2):e54562.
- Line numbers should be provided to help during revision.
Author Response
Reviewer 2
This manuscript is a narrative review to evaluate the impact of sleep-disordered breathing (SBD) on the quality of life and prognosis of patients with idiopathic pulmonary fibrosis (IPF). Among these disorders, obstructive sleep apnea (OSA) and nocturnal hypoxemia (NH) are the most prevalent and extensively studied, whereas central sleep apnea (CSA) has only been documented in recent research. This review provides an overview of the pathophysiological correlations between SBDs and IPF, as well as the currently available therapeutic options, emphasizing the need for further research to optimize clinical management strategies.
It is an interesting work.
It presents excellent figures to show their findings.
I recommend its publication after some minor improvements:
- The authors should describe what this manuscript adds to previous reviews already published: ref 3, 4, etc
- Comment on: Wei CR, Jalali I, Singh J, et al. Exploring the Prevalence and Characteristics of Obstructive Sleep Apnea Among Idiopathic Pulmonary Fibrosis Patients: A Systematic Review and Meta-Analysis. Cureus. 2024;16(2):e54562.
- Line numbers should be provided to help during revision.
Response to Reviewer 2
We sincerely thank the Reviewer for the appreciation of our work and we have highly welcomed all his/her comments on our study and modified the manuscript accordingly.
Reviewer 3 Report
Comments and Suggestions for Authors
- Include a term like "a review" in the title to make it obvious that this is a review article.
- Provide a summary description of the literature search procedure in abstract (e.g., PubMed, Scopus…etc.). Please also give a succinct summary of the main results and relate them to possible effects on lung health.
- Clearly state the research gap at the end of introduction. What is new? Why this paper is important in light of previous reviews (Chronobiology in Medicine 2024;6(3):116-120; Sleep Med Rev. 2016 Apr:26:57-63; Mediators Inflamm. 2015 Apr 5;2015:510105).
- Some references related to the topic are missing (Life (Basel). 2021 Sep 15;11(9):973; J Bras Pneumol . 2024 Nov 16;50(5):e20240058; J Clin Sleep Med . 2021 Jun 1;17(6):1325; Cureus. 2024 Feb 20;16(2):e54562; Experimental and Therapeutic Medicine 29.1 (2025): 16. Pulm Ther 9, 223–236. 2023).
- The lack of methodology used in the selection of the papers given in the review is a crucial problem. A methodology is needed for both the selection of research for the review and the evaluation of the evidence presented in the published studies.
- Table 1 should have a column reflecting the research' limitations.
- Additional tables summarizing the findings from studies in other sections would also be beneficial.
- Please add a list of acronyms at the end.
- Please follow the journal guidelines for references.
Author Response
Reviewer 3
We would like to thank the Reviewer for the accurate reviewing process of our manuscript and we have revised the manuscript according to his/her comments. In particularly:
- Include a term like "a review" in the title to make it obvious that this is a review article.
Response to Reviewer 3’s comment 1
We thank the reviewer for her/his comment and we sincerely apologize for this omission. We
have taken corrective measures by incorporating the term “ review” into the title of our article.
- Provide a summary description of the literature search procedure in abstract (e.g., PubMed, Scopus…etc.). Please also give a succinct summary of the main results and relate them to possible effects on lung health.
Response to Reviewer 3’s comment 2
We sincerely thank the reviewer for his/her interesting comment. As suggested by the reviewer,
we have included in abstract both the search engine used and a summary of our results
- Clearly state the research gap at the end of introduction. What is new? Why this paper is important in light of previous reviews (Chronobiology in Medicine 2024;6(3):116-120; Sleep Med Rev. 2016 Apr:26:57-63; Mediators Inflamm. 2015 Apr 5;2015:510105).
Response to Reviewer 3’s comment 3
We thank the reviewer for this helpful suggestion. We have sought to emphasize the novel
aspects that our review offers compared to previous similar articles as suggested.
- Some references related to the topic are missing (Life (Basel). 2021 Sep 15;11(9):973; J Bras Pneumol . 2024 Nov 16;50(5):e20240058; J Clin Sleep Med . 2021 Jun 1;17(6):1325; Cureus. 2024 Feb 20;16(2):e54562; Experimental and Therapeutic Medicine 29.1 (2025): 16. Pulm Ther 9, 223–236. 2023).
Response to Reviewer 3’s comment 4
We thank the reviewer for his/her comment. As suggested , we have included the articles referenced in our manuscript.
- The lack of methodology used in the selection of the papers given in the review is a crucial problem. A methodology is needed for both the selection of research for the review and the evaluation of the evidence presented in the published studies.
Response to Reviewer 3’s comment 5
We appreciate the reviewers feedback and we would like to provide some clarifications regarding our review. This review is not systematic but rather a narrative review, primarily focused on the pathophysiology of sleep disorders in patients with IPF, with additional consideration given to potential therapeutic options. Due to the limited number of studies in this patient cohort, we have included most of the relevant studies from recent years. The section on biomarkers was included to complement the topic.
- Table 1 should have a column reflecting the research' limitations.
Response to Reviewer 3’s comment 6
We thank the reviewer for her/his comment. In the revised version of the manuscript, table1 includes a column outlining the limitations of each study.
- Additional tables summarizing the findings from studies in other sections would also be beneficial.
Response to Reviewer 3’s comment 7
We thank the reviewer for his/her suggestion. We have added an additional table in the CPAP therapy section.
- Please add a list of acronyms at the end.
Response to Reviewer 3’s comment 8
We thank the reviewer for her/his comment,we have included the list of abbreviations at the
end.
- Please follow the journal guidelines for references.
Response to Reviewer 3’s comment 9
We apologize for the improper format of the references. We have corrected the bibliography according to the guidelines
Round 2
Reviewer 3 Report
Comments and Suggestions for Authors
No further comments.